# Single-atom Ni-N$_4$ provides a robust cellular NO sensor

Min Zhou[1,2,7], Ying Jiang [3,7], Guo Wang[1], Wenjie Wu[2,4], Wenxing Chen [5], Ping Yu [2,6], Yuqing Lin [1✉], Junjie Mao[4✉] & Lanqun Mao [2,6✉]

Nitric oxide (NO) has been implicated in a variety of physiological and pathological processes. Monitoring cellular levels of NO requires a sensor to feature adequate sensitivity, transient recording ability and biocompatibility. Herein we report a single-atom catalysts (SACs)-based electrochemical sensor for the detection of NO in live cellular environment. The system employs nickel single atoms anchored on N-doped hollow carbon spheres (Ni SACs/N-C) that act as an excellent catalyst for electrochemical oxidation of NO. Notably, Ni SACs/N-C shows superior electrocatalytic performance to the commonly used Ni based nanomaterials, attributing from the greatly reduced Gibbs free energy that are required for Ni SACs/N-C in activating NO oxidation. Moreover, Ni SACs-based flexible and stretchable sensor shows high biocompatibility and low nanomolar sensitivity, enabling the real-time monitoring of NO release from cells upon drug and stretch stimulation. Our results demonstrate a promising means of using SACs for electrochemical sensing applications.

[1] Department of Chemistry, Capital Normal University, Beijing 100048, China. [2] Beijing National Laboratory for Molecular Sciences, Key Laboratory of Analytical Chemistry for Living Biosystems, Institute of Chemistry, the Chinese Academy of Sciences (CAS), Beijing 100190, China. [3] College of Chemistry, Beijing Normal University, Beijing 100875, China. [4] Key Laboratory of Functional Molecular Solids, Ministry of Education, Anhui Key Laboratory of Molecule-Based Materials, College of Chemistry and Materials Science, Anhui Normal University, Wuhu 241000, China. [5] Beijing Key Laboratory of Construction Tailorable Advanced Functional Materials and Green Applications, School of Materials Science and Engineering, Beijing Institute of Technology, Beijing 100081, China. [6] School of Chemical Sciences, University of Chinese Academy of Sciences, Beijing 100049, China. [7] These authors contributed equally: Min Zhou, Ying Jiang. ✉email: linyuqing@cnu.edu.cn; maochem@ahnu.edu.cn; lqmao@iccas.ac.cn

Nitric oxide, as one of the most important signaling molecule endogenously generated by cells, functions within the cell or penetrates cell membranes to affect adjacent cells. NO plays major roles in multiple physiological and pathological processes, with the controlled release essential to maintaining vascular homeostasis[1–5]. An overproduction of NO in endothelial cells resulted from exposing mechanical forces, for example, circumferential stretch or fluid shear stress, triggers cascading biological reactions that dysregulates oxidative homeostasis, and causes a range of diseases, including neurodegenerative diseases, autoimmune processes and cancer[6]. Consequently, there has been enormous interest in the development of real-time sensing platforms for NO in normal and pathological conditions, upon mechanical stimulation of cells. Currently, flexible electrochemical sensors featuring excellent stretchability and conductivity have demonstrated to be promising tools to realize real-time monitoring of chemical signals from cells or tissues induced by deformation[7,8]. These sensors usually composed of nanomaterials as electrode materials and flexible polydimethylsiloxane (PDMS) substrate, yielding hybrid scaffold with synergetic functions and better electrochemical properties[9–11]. However, their practical applications are greatly limited by the requirement of noble metal components for efficient catalytic reactions, arguable catalytic mechanism and complicated preparation processes to maximize the utilization of the catalytic sites. Moreover, the transient nature of NO poses additional design challenges of sensing.

Single-atom catalysts (SACs), an emerging heterogeneous material with isolated metal atoms dispersed on solid supports, have received extensive interests since the early demonstration by Zhang et al.[12–14]. Owing to its optimal atom utilization, SACs offer a unique opportunity to rationally design catalysts with excellent activity, selectivity and stability compared with the state-of-the-art nanoparticles based catalysts[15,16]. Aided by significant progresses in synthetic, characterization, and computational modeling techniques, a substantial body of SACs-based research was centered on their application in heterogeneous catalysis field, ranging from the energy conversion to the environmental protection[17–19]. Surprisingly, the biological application of SACs has relatively been less exploited, probably owing to the challenges in controlling over the catalytic activities and biocompatibility of SACs in complexed biological environment. Recently, SACs have been demonstrated to exhibiting excellent performance in enzyme-like catalysis, electrochemical/chemiluminescence sensing and tumor therapy[20–28], highlighting its great potency in applications in natural living system.

Herein, we present the SACs-based electrochemical sensor for the detection of NO in live cellular environment. The system features strategically designed nickel single atoms anchored on N-doped hollow carbon spheres that act as an excellent catalyst for electrochemical oxidation of NO. Notably, Ni SACs/N-C shows superior electrocatalytic performance to the commonly used Ni based nanomaterials, attributing mainly from the greatly reduced Gibbs free energy that are required for Ni SACs in activating NO, a mechanism supported by the density functional theory calculation. Importantly, the Ni SACs/N-C based flexible and stretchable electrochemical sensor shows high biocompatibility, reproducible mechanical compliance, and low nanomolar sensitivity, enabling real-time monitoring of trace amount of NO release from endothelial cells upon drug and stretch stimulation. As far as we know, the development of an easy, stable and even quantitative platform to analyze chemical signals in the environment of living cells has never been achieved before using SACs. This study provides important insights into the design of SACs-based electrochemical platform for cellular sensing applications, and thus broadens the practical applications of SACs.

## Results

**Synthesis and characterization of Ni SACs/N-C.** The nickel single atoms anchored on N-doped hollow carbon spheres were designed to provide a large surface area and good electrical conductivity. As depicted in Fig. 1a, the synthesis of Ni SACs/N-C consists of several steps. Briefly, SiO$_2$ sphere as a template was dispersed in the mixed solution of dopamine hydrochloride and Ni(acac)$_2$. After stirring, polydopamine and nickel acetylacetonate were coated on the surface of the SiO$_2$ sphere to form a core-shell SiO$_2$@polydopamine/Ni(acac)$_2$ structure. The structure was then pyrolyzed at a high temperature in an inert atmosphere and etched with sodium hydroxide to remove SiO$_2$ sphere template. By careful controlling of the Ni weight percent and pyrolysis temperature, Ni SACs/N-C nanospheres can be successfully obtained.

The aberration-corrected high-angle annular dark-field scanning transmission electron microscopy (HAADF-STEM) (Fig. 1b) and transmission electron microscopy (TEM) images (Fig. 1c, d) of Ni SACs/N-C revealed the hollow spherical structures of Ni SACs/N-C, which remain the structure of the hollow N-doped porous carbon spheres (N-C), as shown in Supplementary Fig. 1. No nanoparticles were observed among the carbon spheres, indicating that the atomic Ni sites may be embedded uniformly in the whole carbon spheres. To gain more structural information of the as-synthesized Ni atoms, spherical aberration correction scanning transmission electron microscope (AC-STEM) was performed. Ni atoms with a high atomic number are shown as bright dots (Fig. 1e, f) with atomic dispersed ones highlighted by red circles. Figure 1g shows the energy dispersive X-ray spectroscopy (EDX) elemental mapping of the Ni SACs/N-C, where elements C, N, and Ni were homogeneously distributed over the entire architecture. No diffraction peak of Ni was observed in the X-ray powder diffraction (XRD) pattern for the Ni SACs/N-C (Supplementary Fig. 2), indicating that Ni atoms were anchored on the carbon spheres with even distribution. The binding spectrum of nitrogen was measured by XPS (Supplementary Fig. 3a), which revealed the coexistence of pyridinic (398.4 eV), pyrrolic (400.4 eV) and graphitic (401.4 eV) nitrogen species. The pyrrole-N species is the main anchoring point for stabilizing single atomic Ni because of the strong coordination affinity. For the C1s spectrum in Supplementary Fig. 3b, three peaks with binding energies at 284.5, 286.3, and 287.9 eV were attributed to the graphitic C, C–O and C=O, respectively, indicating the oxygen-containing property of the Ni SACs/N-C. The Ni nanoparticle immobilized on N-doped porous carbon spheres material (Ni NPs/N-C) was fabricated as control material. As shown in Supplementary Figs. 4 and 5, Ni NPs/N-C was made up of Ni, N, C, O elements and formed by random aggregation of nickel atoms during pyrolysis because exogenous nickel atoms cannot bind with nitrogen.

To further explore the electronic structure and coordination environment of Ni species, X-ray absorption near-edge structure (XANES) and extended X-ray absorption fine structure (EXAFS) spectroscopy were performed. Figure 2a shows the XANES curves of Ni K-edge for Ni SACs/N-C, with NiO and Ni foil as references. The position of absorption edge of Ni SACs/N-C was located between those of Ni foil and NiO, indicating that the valence state of Ni in Ni SACs/N-C was between Ni$^0$ and Ni$^{II}$. Fourier transform (FT) EXAFS of Ni SACs/N-C was shown in Fig. 2b. Compared with Ni foil and NiO, Ni SACs/N-C exhibited only one dominant peak at 1.32 Å, which can be attributed to the first coordination shell of Ni-N. No obvious Ni-Ni peaks in the FT-EXAFS spectrum of Ni SACs/N-C were observed, revealing the atomic dispersion of Ni in the hollow carbon support. As shown in Fig. 2c and Supplementary Table 1, the quantitative structural parameters of Ni in Ni SACs/N-C were obtained by

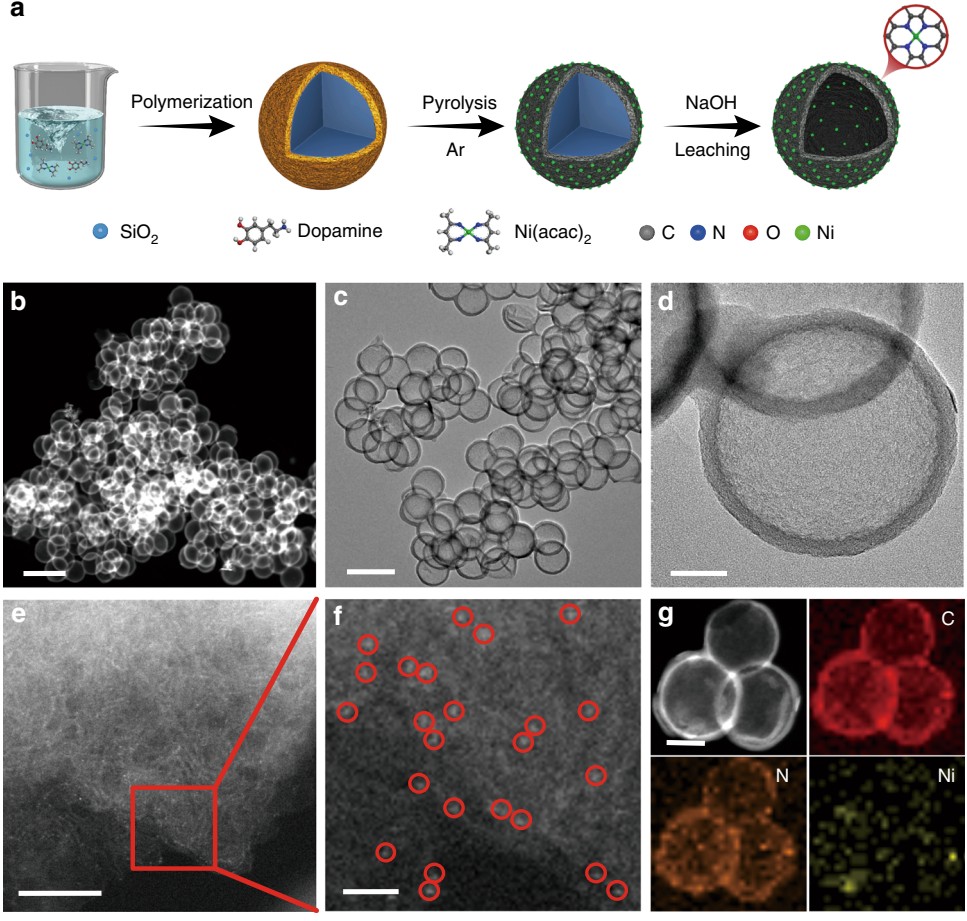

**Fig. 1 Characterization of Ni SACs/N-C. a** Schematic illustration of the synthesis of Ni SACs/N-C. **b** HAADF-STEM image of Ni SACs/N-C. Scale bar: 500 nm. **c, d** TEM images of Ni SACs/N-C. Scale bar: 300 nm in (c), 50 nm in (**d**). **e, f** AC-STEM images of Ni SACs/N-C. Ni single atoms were indicated by red circles. Scale bar: 5 nm in (e), 1 nm in (f). **g** HAADF-STEM image and corresponding EDX element mapping of the Ni SACs/N-C: C (red), N (orange) and Ni (yellow). Scale bar: 100 nm.

least-squares EXAFS fitting. The analysis shows that the coordination number of Ni was approximately 4, and the average bond length of $Ni-N_4$ is 1.92 Å. Based on the above analysis, the local atomic structure around Ni can be constructed, proving that the isolated Ni atom is tetra-coordinated by N atom to form the $Ni-N_4$ structure that is anchored in the nitrogen-doped porous carbon matrix (Supplementary Table 2). The most stable coordination structure could also be determined by DFT calculation[29–31]. To further confirm the geometrical aspects of Ni SACs/N-C prepared in this study, several models including graphene, a (6,6) carbon nanotube, a (10,0) carbon nanotube, $C_{60}$ and phenanthroline with different curvatures were proposed as carbon substrate for the $Ni-N_4$ structure. Theoretically calculated spectra are almost consistent with the experimental ones. It is likely that the models proposed with the carbon structures mentioned above as the substrate for the $Ni-N_4$ structure may all exist in the experiments, of which the most possible one is the graphene model, as shown in Fig. 2d and Supplementary Fig. 6.

**Electrochemical performance and mechanistic study of Ni SACs/N-C-catalyzed NO oxidation.** The electrochemical oxidation of NO is a promising technology for exploring the multiple roles of NO in biological system, in which hunting for effective catalysts for NO oxidation is the main challenge for highly sensitive and efficient determination of NO. Having prepared and characterized the atomically dispersed Ni SACs/N-C, we then set

out to study its electrocatalytic ability towards NO oxidation. To this end, we modified the catalyst on a glassy carbon electrode for all the subsequent testing. Electrochemical oxidation of NO on Ni SACs/N-C was measured by cyclic voltammetry (CV), as shown in Fig. 3a. Ni SACs/N-C exhibits excellent activity towards NO oxidation that commences at ca. +0.60 V and researches a well-defined oxidation peak at +0.83 V (red curve). These potentials are more negative than those at Ni NPs/N-C (blue curve) and N-C (green curve), demonstrating the higher catalytic activity of Ni SACs/N-C. Moreover, the peak current for NO oxidation at Ni SACs/N-C was much higher than those at Ni NPs/N-C and N-C, as shown in Fig. 3b. It's worth noting that in these experiments, the amounts of Ni (by weight) modified onto the electrode in Ni SACs/N-C and Ni NPs/N-C were controlled to be the same. Therefore, the observed better catalytic performance of NO oxidation from Ni SACs/N-C was most probably resulted from the atomically dispersed catalytic sites. Ni SACs/N-C has higher turnover frequency (TOF) value of $1.23 \times 10^4 \, h^{-1}$ than Ni NPs/N-C at the applied potential of +0.85 V (Supplementary Fig. 7a), again suggesting the better catalytic performance toward NO oxidation of Ni SACs/N-C. The Ni SACs/N-C based sensor also has a good durability (Supplementary Fig. 7b), validating its application for continuous sensing of NO. In addition, we investigated the NO adsorption behavior on the SACs/N-C using Fourier-transform infrared (FTIR) spectroscopy and found that the adsorption of NO on Ni site belongs to the top but not to the bridged type (Supplementary Fig. 7c), implying that the

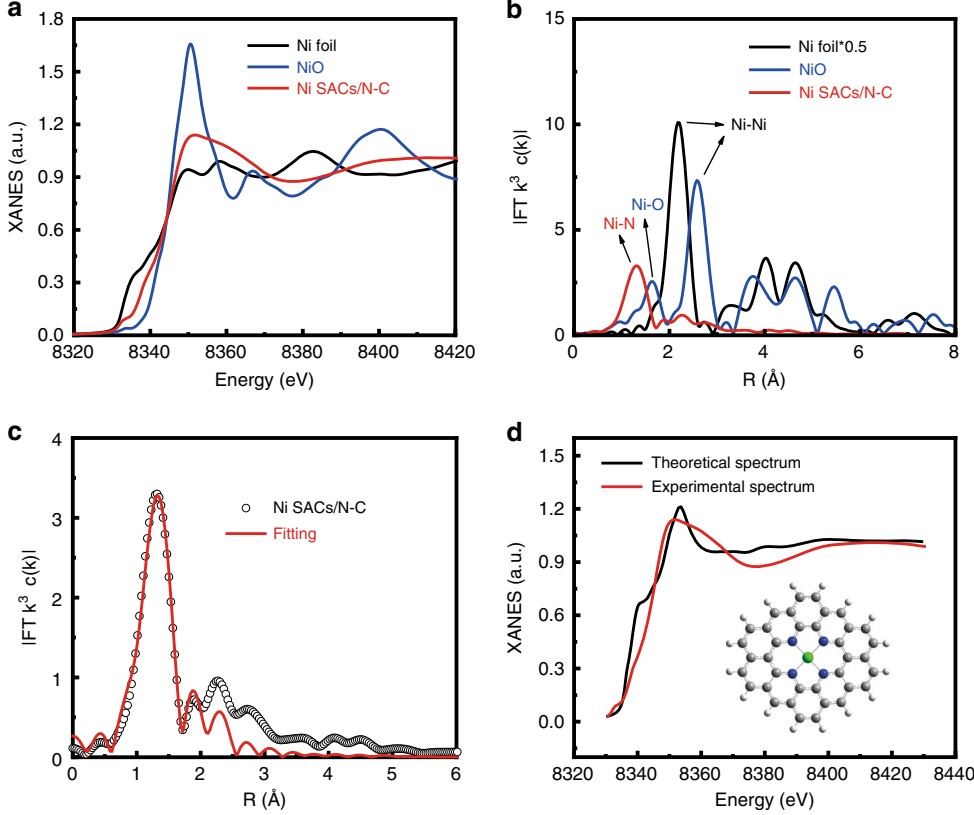

**Fig. 2 Structural analysis of Ni SACs/N-C. a** XANES spectra of Ni SACs/N-C. **b** Fourier transforms of the EXAFS spectra for the Ni K-edge of Ni SACs/N-C, NiO and Ni foil. **c** The corresponding EXAFS fitting curves of the Ni SACs/N-C nanospheres at R space. **d** Comparison between the XANES experimental spectrum for graphene models of Ni SACs/N-C at Ni K edge (red curve) and the theoretical spectra calculated with the depicted structures (black curve). Inset, proposed Ni-N$_4$ architectures.

single-atom state of Ni was maintained after NO adsorption. We also found that the modification of Nafion on the sensor surface well suppressed the current response toward nitrite (NO$_2^-$) (Supplementary Fig. 8), which acts as the main interference toward NO detection[32].

Having demonstrated the excellent catalytic activity of Ni SACs for the electrooxidation of NO, we moved on further to perform mechanistic study by using DFT calculation. It is generally considered that in a typical catalytic reaction, NO molecule first loses one electron to form NO$^+$, as reported previously[9]. However, this is an energetically unfavorable step since the first ionization energy of NO molecule is higher than 9 eV. Here, we proposed the following mechanism (Eqs. (1) and (2) to describe the electrocatalytic reaction, in which the Eq. (1) refers to reaction from step 1, whereas Eq. (2) refers a total reaction from consecutive step 2 to 5 as shown in Fig. 3c. The detailed mechanism including all the proposed reactions were included and discussed in Supplementary Fig. 9,

$$^* + NO =\, ^*NO \tag{1}$$

$$^*NO + H_2O =\, ^* + HNO_2 + H^+ + e^- \tag{2}$$

The * stands for the active sites in SACs, while *NO represents the catalyst with NO molecule adsorbed. The changes in the Gibbs free energy (including the zero-point energy) in Eqs. (1) and (2) on Ni SACs/N-C (0.67 and −0.13 eV, Fig. 3d, black curve) and on Ni NPs/N-C (−1.18 and 1.72 eV, Fig. 3d, red curve) were calculated, which were shown in Fig. 3d and also demonstrated in Supplementary Fig. 9. We also considered the solvation effect with the Polarized Continuum Model (PCM). We

found that the change of Gibbs free energy of Eq. (2) only increases by 0.08 eV, and that of Eq. (1) remains almost unchanged, both showing the minimal effect of solvation. The Ni atom has a 3d$^8$4s$^2$ electronic configuration, whereas its 3d and 4p orbitals are fully occupied by the four ligands in the Ni SACs/N-C. Thus, further coordination with a NO molecule is less favorable than that for Ni NPs/N-C. Since the change of Gibbs free energy for the total reaction NO + H$_2$O = HNO$_2$ + H$^+$ + e$^-$ is fixed, Eq. (2) is favorable for the Ni SACs/N-C, therefore reasonably explaining the excellent electrocatalytic property of Ni SACs/N-C towards NO oxidation as described above.

**Real-time sensing of NO release with the Ni SACs/N-C-based stretchable sensor.** To realize real-time sensing of NO release from cells under mechanical deformation, Ni SACs/N-C was confined on a flexible PDMS substrate to form a stretchable electrochemical sensor (Fig. 4a). Ni SACs/N-C used here features both high catalytic activity and excellent electrical conductivity for NO oxidation, simplifying the sensor preparation. This is remarkable because most of the stretchable sensors reported so far have normally been prepared with composite materials respectively in charge of catalytic recognition of target and electric conducting[10,11]. As shown in Fig. 4b, the as-prepared Ni SACs/N-C-based stretchable electrochemical sensor is well responsive to NO with a nanomolar detection limit. The sensitivity is 430.6 nA μM$^{-1}$ cm$^{-2}$, which is higher than those of the stretchable electrochemical sensors prepared with single metal nanomaterials[7,33], carbon materials[34], and even composite electrode materials[11,35–37] (Supplementary Table 3). We also studied the selectivity of the Ni SACs/N-C based sensor against

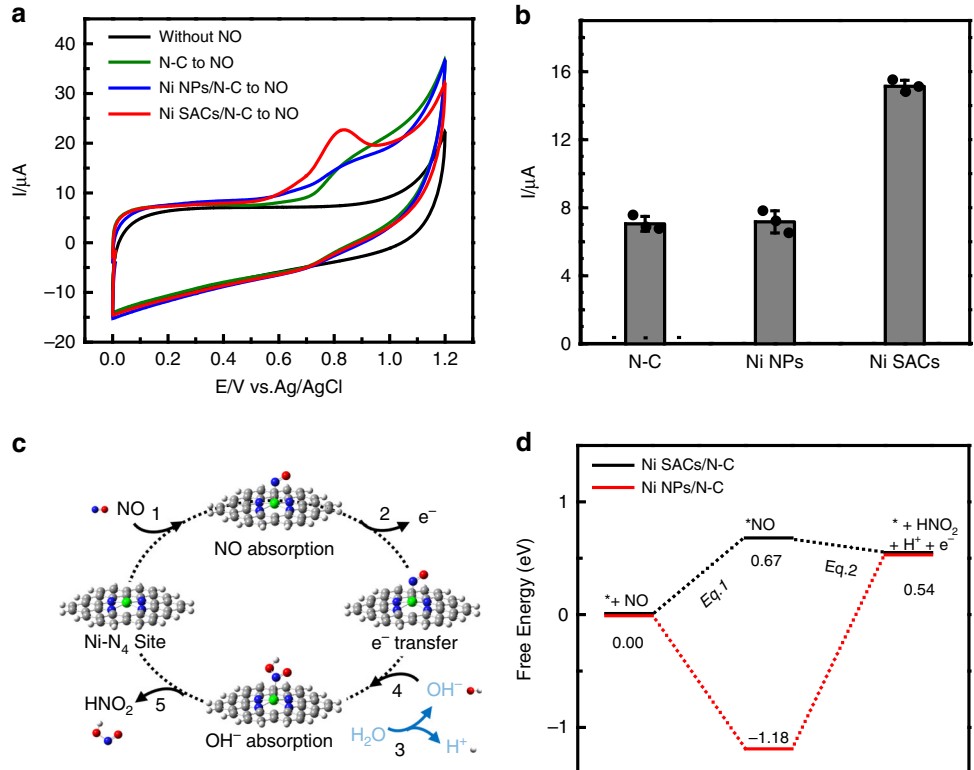

**Fig. 3 Performance and mechanism of Ni SACs/N-C for electrocatalytic NO oxidation. a** Cyclic voltammograms obtained at glassy carbon electrodes modified with Ni SACs/N-C (red curve), Ni NPs/N-C (blue curve) or N-C (green curve) in deaerated phosphate–buffered solution (PBS) containing 0.18 mM NO. Black curve represents a control experiment of Ni SACs/N-C in deaerated PBS without NO. **b** Data are represented as current response recorded with three different modified electrodes shown in (a). Error bars=standard deviation ($n = 3$). **c** The proposed structures for electrocatalytic oxidation process of NO on Ni SACs/N-C. **d** The Gibbs free energy profile along the pathway from NO to HNO$_2$.

commonly interfering species in a biological system, and found that that the sensor has excellent selectivity toward NO sensing (Supplementary Fig. 10). Figure 4c demonstrates the calibration curves of the sensor toward the successive addition of NO before and after 50% stretching the sensor. The similar calibration curves were obtained, demonstrating the high tolerance to mechanical stretching of our sensor. Further tests showed that our sensor has a long-term resistance stability against mechanical bending (Supplementary Fig. 11). When the sensor was bent to a different curvature (Supplementary Fig. 11a) or for 1000 cycles with a radius of 5 mm (Supplementary Fig. 11b), the change of deformed resistance/original resistance (R/R$_0$) remained almost negligible, showing the mechanical stability of the Ni SACs/N-C-based sensor.

Next, we explored whether the Ni SACs/N-C-based stretchable sensor could be used to detect NO in the cellular environment, with human umbilical vein endothelial cells (HUVECs) as a model cell line. HUVECs were cultured on the surface of the stretchable sensor for 24 h, reaching a density of $2 \times 10^6$ cells/cm$^2$. Most of the cells maintained their regular shapes (Fig. 4d), showing the excellent biocompatibility of Ni SACs/N-C under the conditions employed here. Further toxicity tests showed that more than 80% of HUVECs maintain cell viability after co-cultured with different concentrations of Ni SACs/N-C for 48 h (Supplementary Fig. 12). We then studied the capability of the Ni SACs/N-C-based sensor to monitor NO release from HUVECs under stimulation of L-arginine (L-Arg). It is known that L-Arg treatment activates the nitric oxide synthase (NOS) and induces the production of NO in endothelial cells[38–40]. As shown in Fig. 4e (black curve), the treatment of 10 mM of L-Arg led to an obvious increase of the oxidation current. Moreover, when we

simultaneously treated HUVECs with 10 mM L-NG-Nitroarginine Methyl Ester (L-NAME), a precursor of nitric oxide synthase (NOS) inhibitor, and 10 mM L-Arg, no current increase was observed (Fig. 4e, red curve), confirming the current response of the Ni SACs/N-C is due to the release of NO. In addition, the introduction of L-Arg solution (10 mM) to the Ni SACs/N-C-based sensor without cells generated no perceptible current response (Fig. 4e, blue curve). Taken together, these results substantially demonstrate that the Ni SACs/N-C-based stretchable sensor could selectively and sensitively detect the dynamics of NO in the cellular environment.

To further demonstrate the application of the stretchable electrochemical sensor prepared with Ni SACs/N-C, the sensor incubated with HUVECs on the surface was mounted on a special bracket capable of applying precisely controlled mechanical force. When the sensor was stretched to approximately a 50% change in the electrode area, an increase in the current signal (i.e., 520 nA) was observed (Fig. 4g, black curve) and the corresponding schematic cellular mechanotransduction and NO oxidation mechanism was proposed in Fig. 4f. In addition, much smaller current change (ca. 10 nA) was recorded on the Ni SACs/N-C-based stretchable sensor without cell at 50% deformation (Fig. 4g, blue curve), which may be ascribed to the capacitive current change, highlighting that the mechanical force itself contributes minimally to the current response recorded with cells on the sensor surface. Moreover, when cells were pretreated with L-NAME (which inhibits the generation of NO), followed by 50% mechanical force stimulation, less than 10% of the current response was observed (Fig. 4g, red curve), as compared with the groups without L-NAME treatment (Fig. 4g, black curve). It is of note that an extensive mechanical strain not only provokes

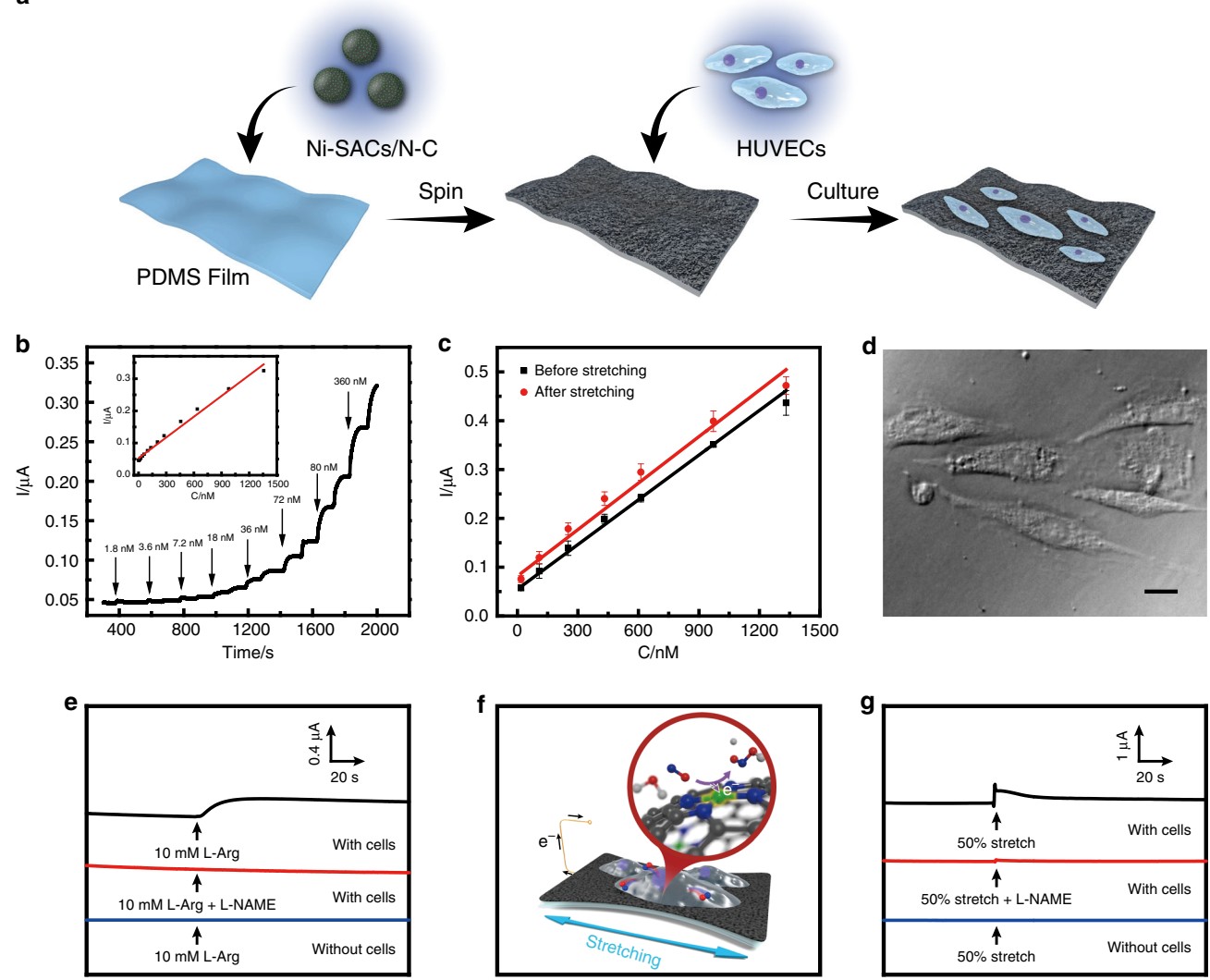

**Fig. 4 Real-time detection of NO released from living cells. a** Schematic illustration of the fabrication of Ni SACs/N-C-based stretchable sensor for NO sensing and HUVECs culturing. **b** Amperometric response of the Ni SACs/N-C-based stretchable sensor (0.5 cm$^2$) to successive addition of NO at +0.80 V. Inset, calibration curve. **c** Calibration curves of the Ni SACs/N-C-based sensor for NO before (black curve) and after (red curve) sensor stretching. Error bars = standard deviation ($n = 3$). **d** Microscopic image of HUVECs cultured on the surface of the Ni SACs/N-C-based sensor for 24 h. Scale bar: 20 µm. **e** Monitoring NO release from HUVECs subject to 10 mM L-Arg with the Ni SACs/N-C-based sensor (black curve). Red and blue curves represent the current responses under stimulation of 10 mM L-Arg and L-NAME (red curve) to cells and 10 mM L-Arg stimulation without cells (blue curve). **f** Diagram of sensor stretching, and **g** real-time monitoring of NO released from HUVECs subject to 50% stretch moduli (black curve) with the Ni SACs/N-C-based sensor. Red and blue curves represent the current responses from the sensor under 50% stretch and L-NAME stimulation with cells (red curve) and under 50% stretch stimulation without cells (blue curve).

HUVECs to release of NO, but also triggers other signaling pathways[41–43]. Taken together, these results suggest that Ni SACs/N-C with atomically dispersed Ni-N$_4$ catalytic site presents a robust electrocatalyst for NO oxidation and sensing, making it reliable as a stretchable electrochemical sensing material in biological detection.

## Discussion

In summary, by using nickel single-atom based electrocatalyst, we demonstrate that the SACs facilities NO oxidation and accomplish the real-time NO sensing in live cellular environment. DFT calculation reveals a two-step catalytic reaction mechanism involved in activating NO with Ni SACs/N-C, in which a greatly reduced Gibbs free energy process enables Ni SACs/N-C with high electrocatalytic performance. Moreover, when the catalytic activity of the Ni SACs/N-C was integrated with a flexible sensor

platform, a highly robust, biocompatible and low-nanomolar sensitive electrochemical tool was created, allowing the recording of trace amount of NO release under a physiological environment. The designing of this single-atom nickel catalyst not only provides great opportunities for live cell analysis but also sheds light on our understanding of electrocatalytic oxidation of NO through theoretical insights, opening a paradigm for the development of SACs-based advanced sensing device for potential healthcare monitoring.

## Methods

**Preparation of N-C.** The preparation of different sized silica spheres involves the ammonia-catalyzed hydrolysis and condensation of tetraethyl orthosilicate (TEOS) in an aqueous ethanol solution via the classical stöber method[44]. To do this, 100 mL of absolute ethanol, 30 mL of deionized water, and 5 mL of 28% NH$_3$ H$_2$O were mixed and stirred. A total of 5 mL of TEOS was added into the mixture and stirred for 1 h. Then 10 mL mixture of ethanol and water containing of 0.5 g dopamine

monomers was added into the preceding mixture and stirred. After reaction for 12 h, the silica spheres were isolated by centrifugation, dried at 70 °C under vacuum, heated to 900 °C for 3 h at the heating rate of 3 °C/min under flowing of Ar gas, and finally naturally cooled to room temperature. The obtained samples were etched in the aqueous solution of 4 M sodium hydroxide for 24 h and collected by centrifugation, washed subsequently with water and ethanol and finally dried at 70 °C under vacuum for overnight.

**Preparation of Ni SACs/N-C.** Ni SACs/N-C was synthesized with the procedures similar to those for the synthesis of N-C. Briefly, 100 mL of absolute ethanol, 30 mL of deionized water, and 5 mL of 28% NH$_3$·H$_2$O were mixed and stirred. A total of 5 mL of TEOS was added into the mixture and stirred for 1 h. Then, 10 mL mixture of ethanol and water containing 0.5 g dopamine monomers and 5.62 mg Ni(acac)$_2$ was added into the preceding mixture and stirred. After a reaction time of about 12 h, the silica spheres were isolated by centrifugation, dried at 70 °C under vacuum, heated to 900 °C for 3 h at the heating rate of 3 °C /min under flowing Ar gas, and finally naturally cooled to room temperature. The obtained samples were etched in the aqueous solution of 4 mol/L sodium hydroxide for 24 h and collected by centrifugation, washed subsequently with water and ethanol and finally dried at 70 °C under vacuum for overnight. The Ni content in Ni SACs/N-C determined by inductively coupled plasma optical emission spectrometry (ICP-OES) is ~0.10% (wt%).

**Preparation of Ni NPs/N-C.** A 56.2 mg Ni(acac)$_2$ and 0.1 g dopamine derived hollow carbon sphere were mixed and grinded. The mixture was then heated to 900 °C for 1 h at the heating rate of 3 °C/min under the flowing of Ar gas and naturally cooled to room temperature. The obtained samples were washed with water and ethanol for 2–3 times, collected by centrifugation and finally dried at 70 °C under vacuum for overnight. The Ni content in Ni NPs/N-C determined by ICP-OES is approximately 0.85% (wt%).

**Preparation of Ni SACs/N-C based sensor.** Ni SACs/N-C suspension (2 mg mL$^{-1}$) was prepared by dispersing Ni SACs/N-C into mixture of deionized water and N, N-dimethylformamide solution under ultrasonication to form homogeneous dispersion. PDMS film (~500 μm in thickness) was obtained by spin-coating the degassed liquid prepolymer and cross-linker (w/w = 10:1) on a plastic substrate at a spin rate of 500 rpm for 10 s and thermally cured at 60 °C. Before PDMS was fully cured, Ni SACs/N-C suspension was dropped and dried at 60 °C for 1 h. Then, the Ni SACs/N-C/PDMS film was peeled off from the substrate and contacted with a copper wire via carbon paste. Finally, the Ni SACs/N-C/PDMS film electrode was further coated with Nafion film by dropping 4 μL 0.5% (w/v) Nafion solution in ethanol on the electrode surface (0.5 cm$^2$) and the electrode was dried in air to form Ni SACs/N-C-based sensor.

**Characterization.** TEM images were taken on a Hitachi HT7700 transmission electron microscope. The high-resolution TEM, HAADF-STEM images, and the corresponding energy-dispersive X-ray (EDX) mapping were recorded by a JEOL JEM-2100F high-resolution transmission electron microscope operating at 200 kV. Inductively coupled plasma optical emission spectroscopy (ICP-OES) was measured by Thermo Fisher IRIS Intrepid II. Powder X-ray diffraction patterns were measured with a Bruker D8 with Cu Kα radiation (λ = 1.5406 Å). The X-ray photoelectron spectroscopy (XPS) was measured by a PHI Quantera SXM system under 3.1 × 10$^{-8}$ Pa using Al$^+$ radiation at room temperature.

**XAFS measurements and analysis.** The X-ray absorption find structure spectra Ni K-edge were collected at BL1W1B station in Beijing Synchrotron Radiation Facility (BSRF). The data were collected in fluorescence excitation mode using a Lytle detector. All samples were pelletized as disks of 13 mm diameter using graphite powder as a binder. The Ni L-edge absorption near-edge structure spectra were collected at BL4B9B station in BSRF. The acquired EXAFS data were processed according to the standard procedures using the ATHENA module implemented in the IFEFFIT software packages. The EXAFS spectra were gained by subtracting the post-edge background from the overall absorption and then normalizing with respect to the edge-jump step. Subsequently, the χ(k) data were Fourier transformed to real (R) space using a hanning windows (dk = 1.0 Å$^{-1}$) to separate the EXAFS contributions from different coordination shells. To obtain the quantitative structural parameters around central atoms, least-squares curve parameter fitting was carried out using the ARTEMIS module of IFEFFIT software packages. The following EXAFS equation was applied (Eq. (3)):

$$\chi(k)=\sum_{j}\frac{N_jS_o^2F_j(k)}{kR_j^2}\cdot\exp\left[-2k^2\sigma_j^2\right]\cdot\exp\left[\frac{-2R_j}{\lambda(k)}\right]\cdot\sin\left[2kR_j+\phi_j(k)\right] \quad (3)$$

$S_0^2$ is the amplitude reduction factor, $Fj(k)$ is the effective curved-wave backscattering amplitude, $N_j$ is the number of neighbors in the $j^{th}$ atomic shell, $R_j$ is the distance between the X-ray absorbing central atom and the atoms in the $j^{th}$ atomic shell (backscatter), λ is the mean free path in Å, $\phi_j(k)$ is the phase shift (including the phase shift for each shell and the total central atom phase shift), σj is the Debye-Waller parameter of the jth atomic shell (variation of distances around the

average $R_j$). The functions $F_j(k)$, λ and $\phi_j(k)$ were calculated with the ab initio code FEFF8.2.

**Electrocatalytic oxidation of NO on Ni SACs/N-C.** Briefly, NO was prepared by slowly dropping 4 M H$_2$SO$_4$ into deoxygenated 2 M NaNO$_2$ solution[7]. A 1.8 mM NO-saturated solution was obtained by perfusing pure NO gas into PBS for 30 min at 20 °C. Electrochemical measurements were performed on a computer-controlled electrochemical analyzer (CHI 660, Shanghai, China) with a three-electrode system with the Ni SACs/N-C/PDMS as working electrode, Ag/AgCl as reference electrode and Pt wire as counter electrode. A 0.10 M PBS was used as the electrolyte. All electrochemical experiments were carried out under the protection of nitrogen gas atmosphere. Calibration of Ni SACs/N-C-based sensor for NO detection was performed by adding a series of NO standard solution aliquots into the deaerated PBS. The electrode potential was held at +0.80 V vs. Ag/AgCl for amperometric detection.

The TOF (h$^{-1}$) for NO was calculated as using Eq. (4) as shown below:

$$\text{TOF} = \frac{I_{\text{product}}/NF}{m_{\text{cat}}\times\omega/M_{Ni}}\times 3600 \quad (4)$$

$I_{\text{product}}$: partial current for certain product, NO;
$N$: the number of electron transferred for product formation, which is 1 for NO;
$F$: Faradaic constant, 96485 C mol$^{-1}$;
$m_{\text{cat}}$: catalyst mass in the electrode, g;
ω: Ni loading in the catalyst;
$M_{Ni}$: atomic mass of Ni, 58.69 g mol$^{-1}$.

**Human umbilical vein endothelial cell culture.** Human umbilical vein endothelial cells (HUVECs) were purchased from ATCC (PCS-100-010) and were cultured using Endothelial medium (ScienCell,1001) at 37 °C in a humidified incubator (95% air with 5% CO$_2$). Specially, the medium was made from 500 mL basal medium, 25 mL fetal bovine serum (FBS), 5 mL endothelial cell growth supplement (ECGS) and 5 mL penicillin/streptomycin solution. For cells experiment, HUVECs were cultivated on Ni SACs/N-C-based sensor which was kept in the incubator for 24 h to allow cells to adhesion. These loosely bounded HUVECs were washed with sterile PBS before electrochemically recording of NO release.

**Real-time monitoring of NO release during cell mechano-transduction.** In order to apply stretch strains to HUVECs, the Ni SACs/N-C-based sensor (active area, 0.5 cm$^2$) cultured with HUVECs was carefully held with clamps on the transformable device and different stretching force was exerted on the sensor. The stretching and releasing time course of each strain was 5 s and 0.5 s, respectively. The concentrations of both L-Arg and L-NAME (NOS inhibitor) added to HUVECs culture were 10.0 mM.

**Computational details.** In order to explore the geometrical and catalytic properties of the Ni SACs, several models with different curvatures were proposed for the Ni-N$_4$ structure. These structures were geometrically optimized with the B3LYP hybrid density functional[45,46] by using the Gaussian03 program[47]. The Lanl2DZ pseudopotential for the Ni atoms and 6–31 G (d) basis sets for the other atoms were used in the calculations. All the structures were verified with vibrational analysis and no imaginary frequency was found. The zero-point energies and Gibbs free energies were also obtained in the vibrational analysis. After geometrical optimization, the XANES spectra of the models were calculated by using the ab initio multiple-scattering code JFEFF[48]. Full multiple scattering algorithm was used.

**Reporting summary.** Further information on research design is available in the Nature Research Reporting Summary linked to this article.

## Data availability

The data that support the findings of this study are available from the authors upon reasonable request. The supplementary data are also available at public repositories (Identifier: DOI 10.17605/OSF.IO/7TZNY). The source data underlying Figs. 2a–d, 3a, b, d, and 4b, c, e, g and Supplementary Figs. 2, 3a–c, 5a–c, 6, 7, 8, 10–12 are provided as a Source Data file.

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

## Acknowledgements

This work was supported by the National Natural Science Foundation of China (No. 21575090 for Y.L.; Nos. 21790390 and 21790391 for L.M.; No. 21971002 for J.M.; Nos. 21775151 and 21790053 for P.Y.), the National Key Research and Development Project (Grant No. 2018YFE0200800), the National Basic Research Program of China (Grant Nos. 2016YFA0200104), the Strategic Priority Research Program of Chinese Academy of Sciences (XDB30000000), the Beijing Nova Program of Science and Technology (Z191100001119108), Chinese Academy of Sciences (QYZDJ-SSW-SLH030), High-level Teachers in Beijing Municipal Universities in the Period of 13th Five year Plan (CIT&TCD20190330), Scientific Research Project of Beijing Educational Committee (KM201810028008) and Youth Innovative Research Team of Capital Normal University.

## Author contributions

L.M., Y.L., and J.M. conceived the idea for the project. J.M. and W.W. conducted material synthesis and structural characterizations. W.C. conducted XAFS measurements. M.Z. and Y.J. designed and performed catalytic performance testing, and cell studies. P.Y. and Y.J. analyzed the data; G.W. performed DFT calculations. M.Z., J.M., Y.L., and Y.J. drafted the manuscript, Y.J. and L.M. finalized the manuscript. M.Z. and Y.J. contributed equally to this work. All authors discussed and commented on the manuscript.

## Competing interests

The authors declare no competing interests.
