## [Peer Review File · Nature Communications]

Reviewers' Comments:

Reviewer #1:

Remarks to the Author:

The authors anchored Ni-N4 on N-doped hollow carbon spheres to fabricate metal/carbon nano-composite catalysts they called them Ni SACs/N-C. Their results showed that this kind of catalysts showed superior electrocatalytic performance to the commonly used Ni based nanomaterials toward the electrochemical detection of NO. Further application of this sensor was demonstrated by real-time monitoring of NO release from endothelial cells cultured on Ni SACs/N-C based flexible and stretchable sensor upon drug and stretch stimulation. This is an interesting work since they introduce the single-atom catalysts into biological measurement, but there are some key issues that must be addressed before considering accepting this work for publication.

1. As stated in the manuscript, although it was not designed for NO sensing, the strategies of transition metal (e.g., Fe, Ni, Co, Mn) single-atom catalysts for electrochemical applications have been reported by several works, which were used for oxygen reduction or hydrogen evolution. In this aspect, the innovation of this work was reduced. Therefore, the novelty of this work should be better clarified.

2. I am a little confused about the term of single-atom catalyst. According to the results demonstrated in this work, Ni-N4 was anchored on carbon skeleton structure by C-N bond, in this case, can we call this material single-atom catalyst? Further, there are many kinds of molecules containing Ni-N4 structure (e.g. Ni porphyrin) that show excellent electrochemical catalytic performance, I am wondering if there any difference in terms of the catalytic performance and mechanism between the single-atom catalyst and the Ni-N4 molecules.

3. The authors demonstrated that this catalyst showed superior electrocatalytic performance to the commonly used Ni based nanomaterials toward the electrochemical detection of NO. However, for electrochemical detection of NO, there are numerous sensors showing very excellent sensing performance such as sub-nM detection limit and negatively shifted oxidation peak, while the Ni SACs/N-C only showed nM detection limit and oxidation peak at 0.8 V.

4. I think the selectivity of this sensor should be further demonstrated, and the interferences should be systematically investigated.

5. It was interesting that the Ni SACs/N-C catalysts were used to construct flexible and stretchable sensor. However, given that the material is a hollow spherical structure and would possibly be separated by bending and or stretching manipulation, how to maintain the conductivity and electrochemical stability during these manipulations.

6. For the detection of NO release from stretched HUVECs, have the authors obtained some biologically important phenomena that we did not know before?

7. Have the authors optimized the ratio between dopamine hydrochloride and Ni(acac)₂ for the fabrication of Ni SACs/N-C catalysts?

Reviewer #2:

Remarks to the Author:

The authors report a Ni-N4 single-atom catalyst aiming at providing a strategy/material for real-time and continuous monitoring of NO under the biological environment. The structure of SAC was firstly characterized using various state-of-the-art techniques including HAADF-STEM, TEM and AC-STEM, followed by XANES and EXAFS to determine the electronic structure and coordination environments. Subsequently, the electrochemical performances were measured and the reaction mechanism was studied using DFT. Finally, the performances of real-time monitoring were examined under stretchable conditions on the PDMS substrate.

Utilizing SAC for NO detection under the biological environment is a quite novel interdisciplinary idea, considering its feasibility and effectiveness, I think this manuscript is interesting and the conclusions might be helpful for future studies. However, there are several flaws that should be addressed before the paper can be considered further.

- (1) The existing evidences can only demonstrate that Ni does exist as a single atom before NO detection. However, can the single-atom state of Ni be maintained after NO adsorption? The adsorption behavior of NO adsorption, as well as the dispersion and oxidation state of Ni could be investigated using the in situ FTIR spectra.
- (2) Why the Ni-N4 structure was assigned to the graphene model? From Figure S6 it seems that the carbon nanotubes could also be reasonable candidates. In addition, is it possible that the structures in Figure S6a-c,e are all existed?
- (3) Besides the proposed mechanisms in Figure 3, it could be possible for the NO and H₂O to co-adsorb on the Ni site, followed by a proton transfer reaction. The authors may want to try this possible pathway. In addition, the solvent effect should be considered during the Gibbs free energy calculations.
- (4) It seems that one proton is missing in Figure 3c, where is it? By the way, a reaction pathway should contain the structures of the reactant, all the intermediates, all the transition state structures (very important) that connect each local/global minimum and the product(s). The authors may want to take a look at the Figure 4 in Nature Chemistry 3 (8), 634-641 to learn what a "reaction pathway" is like.
- (5) As a catalyst, how is the turnover frequency and how is the durability of the SAC?
- (6) The toxicity of the SAC should be further studied in cell environment.

Reviewer #3:

Remarks to the Author:

In this manuscript, the authors reported single-atom catalyst-based electrochemical sensors for NO detection. Single-atom catalysts have attracted great interest in different fields. Constructing single-atom catalysts based sensors have been widely reported. The idea is clever and the procedure appears to be quite simple. The description is also relatively clear. Having said that, I just don't think the work is significant enough to merit publication at Nature Communications. At the same time, some critical issues should be addressed carefully, which are listed as follows.

1. For the preparation of Ni NPs/N-C, why hollow carbon spheres were obtained because no silicon precursor was added?
2. The referee does not think the simulating XANES spectrum is satisfactory in comparison to experimental results.
3. As mentioned, Ni SACs/N-C exhibited lower peak potential for NO oxidation, demonstrating higher catalytic activity of Ni SACs/N-C. However, there seems no distinct difference in peak potentials among three catalysts.
4. Since Ni NPs were used as control, why Ni₂ model was adopted for mechanistic study instead of Ni NPs?
5. To demonstrate the advantage of the obtained catalyst, one table involving electrochemical parameters is recommended to add in supporting information to compare.
6. Should interference species be considered because their high oxidation potential?

Replies to Reviewer 1

General Comment: *The authors anchored Ni-N₄ on N-doped hollow carbon spheres to fabricate metal/carbon nano-composite catalysts they called them Ni SACs/N-C. Their results showed that this kind of catalysts showed superior electrocatalytic performance to the commonly used Ni based nanomaterials toward the electrochemical detection of NO. Further application of this sensor was demonstrated by real-time monitoring of NO release from endothelial cells cultured on Ni SACs/N-C based flexible and stretchable sensor upon drug and stretch stimulation. This is an interesting work since they introduce the single-atom catalysts into biological measurement, but there are some key issues that must be addressed before considering accepting this work for publication.*

Q1: *As stated in the manuscript, although it was not designed for NO sensing, the strategies of transition metal (e.g., Fe, Ni, Co, Mn) single-atom catalysts for electrochemical applications have been reported by several works, which were used for oxygen reduction or hydrogen evolution. In this aspect, the innovation of this work was reduced. Therefore, the novelty of this work should be better clarified.*

A1: We thank the reviewer for the great comment. As pointed out by the reviewer, transition-metal single-atom catalysts (SACs) have been used as effective catalysts, but mainly toward the electrochemical reactions of oxygen reduction, carbon dioxide reduction, hydrogen evolution, oxygen evolution and so forth. While the excellent electrocatalytic performance of SACs suggests the great potential applications of SACs in electrochemical sensing, this potentiality remains largely unexplored so far. In this work, we have developed a flexible SAC-based sensing system with high sensitivity and excellent stability for in situ monitoring the release of biologically important signaling molecules, not just simple detection of NO. The finding of the catalytic NO oxidation properties of Ni SACs, along with the detailed mechanistic study to uncover its reaction pathway, offer more insight into designing new catalysts for fundamental research and help in broadening the practical applications of SACs. Most importantly, the development of an easy, stable and even quantitative platform for assessing biological important species to analyze the biomedical signals, movements and the environment of living cells has never been achieved before using SACs. Our Ni SAC system probably represents the very first example in the field. Considering the scientific importance and potential applications, we believe that our approach will push the field forward for designing wearable and noninvasive SAC-based biosensors for potential healthcare monitoring. We have highlighted the novelty of our work in the revised manuscript (Page 3 and 4).

Q2: *I am a little confused about the term of single-atom catalyst. According to the results demonstrated in this work, Ni-N₄ was anchored on carbon structure by C-N bond, in this case, can we call this material single-atom catalyst? Further, there are many kinds of molecules containing Ni-N₄ structure (e.g. Ni porphyrin) that show excellent electrochemical catalytic performance, I am wondering if there any difference in terms of the catalytic performance and mechanism between the single-atom catalyst and the Ni-N₄ molecules.*

A2: We appreciate the comments from the reviewer. Single-atom catalysts (SACs) refers to a catalyst with isolated metal atoms dispersed on solid supports. The active site generally consists of a single metal atom and other atoms on the surface of the support or adjacent functional species. Similar single-atom metal-N₄ structures were reported in the metal-N-C catalyst system, such as Fe-N₄ (*J. Am. Chem. Soc.* 2018, 140, 37, 11594), Ni-N₄ (*Angew. Chem. Int. Ed.* 2018, 57, 15194) and Mn-N₄ (*J. Am. Chem. Soc.* 2019, 141, 12005). In the Ni SACs/N-C material we

synthesized, Ni is an isolated metal center supported on a carbon substrate and does not have a direct valence bond with each other. In contrast, the metal of M-N₄ in metalloporphyrin exists in the form of cation, as Cu²⁺, Zn²⁺, Ni²⁺, and thus probably cannot be called as single atom catalysts.

There are differences in terms of the catalytic performance and mechanism between the single-atom catalyst and the Ni-N₄ molecules. Specifically, Ni SACs/N-C demonstrates a lower detection limit compared with Ni-porphyrin with M-N₄ structure (*Bioelectrochemistry*, 2007, 71, 46). This is likely resulted from the mono-atomically dispersed metal active centers and larger specific surface area and accelerated electron transfer characteristics associated with carbon-based hollow scaffold which effectively improve atomic utilization and catalytic efficiency. Moreover, the mechanism of Ni-N₄ molecules mediated NO oxidation is much more complicated than of Ni SACs/N-C. For example, a five-step reaction mechanism is usually involved in the electrocatalytic oxidation of NO (*Coord. Chem. Rev.* 2010, 254, 2755) for Ni-N₄ molecules. While in our study, a reasonable two-step mechanism was proposed as demonstrated in the revised manuscript (page 9).

Q3: *The authors demonstrated that this catalyst showed superior electrocatalytic performance to the commonly used Ni based nanomaterials toward the electrochemical detection of NO. However, for electrochemical detection of NO, there are numerous sensors showing very excellent sensing performance such as sub-nM detection limit and negatively shifted oxidation peak, while the Ni SACs/N-C only showed nM detection limit and oxidation peak at 0.8 V.*

A3: We agree with the reviewer that there are a few composite electrode materials have been studied for NO sensing, showing excellent performance in detection limit and negatively shifted oxidation peak (*Anal. Chem.* 2019, 92, 1804; *Anal. Chem.* 2018, 90, 4438; *Nat. Commun.* 2013, 4, 2225). Though our sensor shows a moderate sensitivity, the potential and detection limit of our system are comparable to or even greater than most of the carbon materials and metal nanomaterials based sensors (Table S3). We believe these electrochemical parameters could be even largely improved by future optimization of SACs, for example, increasing the catalytic site, and using more conductive substrates.

Q4: *I think the selectivity of this sensor should be further demonstrated, and the interferences should be systematically investigated.*

A4: We thank the reviewer for this suggestion. We tested the selectivity of the Ni SACs/N-C based sensor with potentially interfering species, including H₂O₂, UA, AA, DA, 5-HT, NO₂⁻, Arg, and GSH in a biological relevant system. As shown (Figure S10), our sensor showed a remarkable current response for NO oxidation (10 μM), while the current responses from each of the potentially interfering specie (10 μM) at the same potential are minimal, demonstrating the excellent selectivity of the sensor.

Q5: *It was interesting that the Ni SACs/N-C catalysts were used to construct flexible and stretchable sensor. However, given that the material is a hollow spherical structure and would possibly be separated by bending and or stretching manipulation, how to maintain the conductivity and electrochemical stability during these manipulations.*

A5: We previously developed a simple ultraviolet (UV)-irradiation-assisted technique to fabricate gold nanoparticle integrated stretchable and flexible electrode (*Anal. Chem.* 2018, 90, 7158). Specifically, the strong adhesion ability during PDMS cross linking was used to tightly bond Ni SACs/N-C to the flexible substrate, followed by pre-bent to allow the gap filled with Ni SACs/N-C. With these careful preparation steps, SACs can be densely and continuously coated onto the film, ensuring the conductivity and electrochemical stability during stretching. Figure 4c and Figure S11 show minimal changes in sensitivity and resistance during the strain, demonstrating the excellent stability of the sensor.

Q6: *For the detection of NO release from stretched HUVECs, have the authors obtained some biologically important phenomena that we did not know before?*

A6: Single atoms catalysts has shown fascinating properties and leads to wide range of application, however, its electrochemical sensing application particularly in live cellular environment is largely unexplored. In this work, we focus mainly on whether we could employ the excellent catalytic properties of SACs in developing powerful sensing tools for biomedical research. Therefore, a classic cell model of mechanical transduction where NO was known to actively participate in the process would be reasonable for the initial validation study. It would be interesting to gain novel insights into the biological role of NO with the current system; however, we feel it would be beyond the scope of this already dense manuscript. In the near future, we will use this tool to explore some unknown yet biologically important phenomena of NO.

Q7: *Have the authors optimized the ratio between dopamine hydrochloride and Ni(acac)₂ for the fabrication of Ni SACs/N-C catalysts?*

A7: Yes, we have optimized the ratio between dopamine hydrochloride and Ni(acac)₂ for the fabrication of Ni SACs/N-C catalysts to avoid the yielding of Ni NPs/N-C. We also found that the electrocatalytic performance of the Ni SACs/N-C material to NO will not be significantly improved with the increased nickel content, therefore, the current ratio of 0.5 g dopamine monomers to 5.62 mg Ni(acac)₂ was selected to synthesize Ni SACs/N-C (Page 13).

Replies to Reviewer 2

General Comment: *The authors report a Ni-N₄ single-atom catalyst aiming at providing a strategy/material for real-time and continuous monitoring of NO under the biological environment. The structure of SAC was firstly characterized using various state-of-the-art techniques including HAADF-STEM, TEM and AC-STEM, followed by XANES and EXAFS to determine the electronic structure and coordination environments. Subsequently, the electrochemical performances were measured and the reaction mechanism was studied using DFT. Finally, the performances of real-time monitoring were examined under stretchable conditions on the PDMS substrate.*

Utilizing SAC for NO detection under the biological environment is a quite novel interdisciplinary idea, considering its feasibility and effectiveness, I think this manuscript is interesting and the conclusions might be helpful for future studies. However, there are several flaws that should be addressed before the paper can be considered further.

Q1: *The existing evidences can only demonstrate that Ni does exist as a single atom before NO detection. However,*

can the single-atom state of Ni be maintained after NO adsorption? The adsorption behavior of NO adsorption, as well as the dispersion and oxidation state of Ni could be investigated using the in situ FTIR spectra.

A1: We thank reviewer for this great suggestion. To answer the question from the reviewer, we carried out the in situ FTIR experiment to study the behavior of NO adsorption, as well as the dispersion and oxidation state of Ni accordingly. As shown in Figure S7C, when Ni SACs/N-C was exposed to NO solution, no infrared absorption signal was detected. Upon applying a potential of +0.80 V (*vs.* Ag/AgCl), an obvious absorption peak located at 1829-1842 cm^{-1} was observed, a frequency range that can be attributed to the top adsorption of NO on Ni site (*J. Phys. Chem. C*, 2019, 123, 21588). Moreover, there is no obvious NO bridged-bound at 1650-1600 cm^{-1} , suggesting the absence of adjacent Ni atoms in Ni SACs/N-C (*ACS Catal.* 2015, 5, 3717), and the single-atom state of Ni was maintained after NO adsorption. It is worth of note that when a potential of +0.80 V was removed, the adsorption signal was diminished, again indicating the well maintained single-atom state of Ni after the reaction and stable catalytic property.

Q2: *Why the Ni-N₄ structure was assigned to the graphene model? From Figure S6 it seems that the carbon nanotubes could also be reasonable candidates. In addition, is it possible that the structures in Figure S6a-c,e are all existed?*

A2: We agree with the reviewer. As shown in Figure S6, for (6,6) and (10,0) carbon nanotubes, the shoulder peaks around 8350 eV are more than that observed in the experiment, suggesting the possibility of co-existence of all structures. We have revised our statement to include this information in the revised SI (Figure S6).

Q3: *Besides the proposed mechanisms in Figure 3, it could be possible for the NO and H₂O to co-adsorb on the Ni site, followed by a proton transfer reaction. The authors may want to try this possible pathway. In addition, the solvent effect should be considered during the Gibbs free energy calculations.*

A3: We thank the reviewer for this great suggestion. We attempted to propose a structure, in which both NO and H₂O molecules are co-adsorbed on the Ni SACs/N-C. However, the structure with NO and H₂O at the same side has not been obtained. This is probably because in a normal hexa-coordinated Ni structure, the Ni atom coordinates with four ligands, forming a rigid structure within a plane, and the other two ligands should be at the different sides in order to form a stable octagon. However, NO and H₂O molecules at the different side cannot react with each other. Therefore, it might be less likely for the NO and H₂O to co-adsorb on the Ni site.

The changes in the Gibbs free energy (including the zero-point energy) in the two steps respectively on Ni SACs/N-C (0.67 and -0.13 eV, Figure 3d, black) and on Ni NPs/N-C (-1.18 and 1.72 eV, Figure 3d, red) were calculated, and shown in Figure 3d and Figure S9. We also considered the solvation effect with the PCM (Polarized Continuum Model) method. We found that the change of Gibbs free energy of reaction (2) only slightly increases by 0.08 eV and that of reaction (1) remains almost unchanged, showing the minimal effect of solvation. We have included the solvation effect in the revised manuscript (Page 9).

Q4: *It seems that one proton is missing in Figure 3c, where is it? By the way, a reaction pathway should contain the structures of the reactant, all the intermediates, all the transition state structures (very important) that connect*

each local/global minimum and the product(s). The authors may want to take a look at the Figure 4 in *Nature Chemistry* 3 (8), 634-641 to learn what a "reaction pathway" is like.

A4: We greatly appreciate the suggestion from this reviewer. The missing proton has been included in the revised Figure 3c and d. In addition, according to the excellent paper suggested by the reviewer (*Nat. Chem.* 2011, 3, 634-641), we have revised the reaction pathway by including the structures of the reactant, all the intermediates, all the transition state structures, as suggested by the reviewer.

Q5: *As a catalyst, how is the turnover frequency and how is the durability of the SAC?*

A5: We determined the turnover frequency (TOF) of Ni SACs and compared it to that of Ni NPs/N-C at the same experimental condition in the revised supporting information. As shown in Figure S7a, Ni SACs/N-C shows higher TOF value than that of Ni NPs/N-C at the potentials applied. Particularly, Ni SACs/N-C shows the highest TOF value ($1.23 \times 10^4 \text{ h}^{-1}$) at +0.85 V, demonstrating the superior catalytic activity of Ni SACs/N-C towards NO oxidation.

We have evaluated the durability of Ni SACs/N-C in the revised SI. As shown in Figure S7b, we observed the current increases quickly upon the addition of 1 μM NO and then gradually becomes stable in 400 s. After that, the current remains unchanged for more than 1 h, indicating the applicability of the Ni SACs/N-C in continuous measurements of NO.

Q6: *The toxicity of the SAC should be further studied in cell environment.*

A6: We investigated the toxicity of Ni SACs/N-C by incubating them with human umbilical vein endothelial cells (HUVECs) for 48 hours, following a Cell counting kit-8 (CCK-8) assay. As shown in Figure S12 in the revised SI, there is no significant cytotoxicity at the concentrations tested here. The excellent biocompatibility results are consistent with most of the carbon-based SACs featuring similar composition of carbon scaffold and few metal single atoms (*Angew. Chem. Int. Ed.* 2020, 132, 2585).

Replies to Reviewer 3

General Comment: *In this manuscript, the authors reported single-atom catalyst-based electrochemical sensors for NO detection. Single-atom catalysts have attracted great interest in different fields. Constructing single-atom catalysts based sensors have been widely reported. The idea is clever and the procedure appears to be quite simple. The description is also relatively clear. Having said that, I just don't think the work is significant enough to merit publication at Nature Communications. At the same time, some critical issues should be addressed carefully, which are listed as follows.*

Response: We thank the reviewer for the great comment. We agree with the reviewer that SACs have attracted great interest in different fields, electrocatalysis in particular. While the excellent electrocatalytic performance of SACs suggests the great potential applications of SACs in electrochemical sensing, this potentiality remains largely unexplored so far. The recent efforts on the sensor development with single-atom catalysts have been made more on optical sensing (spectroscopy, electrochemiluminescence and chemiluminescence) but less on electrochemical

sensing. In this work, we have developed a flexible SAC-based electrochemical sensing system with high sensitivity and excellent stability for in situ monitoring of the release of biologically important signaling molecules, not just simple detection of NO. The finding of the catalytic NO oxidation properties of Ni SACs, along with the detailed mechanistic study to uncover its reaction pathway, offer more insight into designing new catalysts for fundamental research and help in broadening the practical applications. Most importantly, the development of an easy, stable and even quantitative platform for assessing biological important species to analyze the biomedical signals, movements and the environment of living cells has never been achieved before using SACs. Our Ni SAC system probably represents the very first example in the field. Considering the scientific importance and potential applications, we believe that our approach will push the field forward for designing wearable and noninvasive SAC-based biosensors for potential healthcare monitoring. We have highlighted the novelty of our work in the revised manuscript (Page 3 and 4).

Q1: *For the preparation of Ni NPs/N-C, why hollow carbon spheres were obtained because no silicon precursor was added?*

A1: We thank reviewer for this comment. The hollow carbon spheres structure of Ni NPs/N-C was most likely derived from the use of pre-made hollow carbon sphere (N-C) as a precursor during the synthesis. We have provided the detailed synthetic information for better understanding in the revised manuscript Page 13.

Q2: *The referee does not think the simulating XANES spectrum is satisfactory in comparison to experimental results.*

A2: We appreciate the comment from this referee. The reason for the differences between calculated XANES spectrum and the experimental result is probably resulted from the specific condition in the experiment and the approximated defined parameters used in the theoretical calculation. Also, it is highly possible that there are more than one structure existing in our study, as discussed in the Figure S6, Page S7.

Q3: *As mentioned, Ni SACs/N-C exhibited lower peak potential for NO oxidation, demonstrating higher catalytic activity of Ni SACs/N-C. However, there seems no distinct difference in peak potentials among three catalysts.*

A3: We thank the reviewer for this comment. As shown in Figure 3A, we compared the catalytic performance of the three catalysts through cyclic voltammetry, and found that for N-C and Ni NPs/N-C modified electrodes, there are no significant NO electrocatalytic oxidation peaks. Ni SACs/N-C exhibits excellent activity towards NO oxidation that commences at ca. +0.60 V and researches a well-defined oxidation peak at +0.83 V (red curve). These potentials are more negative than those at Ni NPs/N-C (blue curve) and N-C (green curve), demonstrating the higher catalytic activity of Ni SACs/N-C.

To better evaluate the catalytic activity of Ni SACs/N-C, we determined and compared the turnover frequency (TOF) of Ni SACs and Ni NPs/N-C, as shown in Figure S7b. It is clear that Ni SACs/N-C shows higher TOF value than that of Ni NPs/N-C at the potentials applied. Particularly, Ni SACs/N-C shows the highest TOF value at 0.85 V, demonstrating the superior catalytic activity of Ni SACs/N-C towards NO oxidation to that of Ni NPs/N-C.

Q4: *Since Ni NPs were used as control, why Ni₂ model was adopted for mechanistic study instead of Ni NPs?*

A4: The Ni NPs/N-C is composed of not only Ni atoms, but also N and C atoms as implicated in Figure S5. They are quite different from the pure metal nanoparticles. Therefore, pure Ni NPs model was not used. Moreover, the detailed structure of Ni NPs/N-C especially the metal core number is still unclear. Considering the similarity of the N and C part in the Ni₂ model to that of single Ni atom model, we think the Ni₂ model is more suitable for comparing the differences between the single atom and non-single atom material. We have included this discussion in the revised Figure S9.

Q5: *To demonstrate the advantage of the obtained catalyst, one table involving electrochemical parameters is recommended to add in supporting information to compare.*

A5: We thank the reviewer for this great suggestion. According to the suggestion, we summarized and compared the electrochemical parameters of the present work with the previously reported relevant studies and added the table in supporting information (Table S3).

Q6: *Should interference species be considered because their high oxidation potential?*

A6: We thank the reviewer for this suggestion. We tested the selectivity of the Ni SACs/N-C based sensor with potentially interfering species, including H₂O₂, UA, AA, DA, 5-HT, NO₂⁻, Arg, and GSH in a biological relevant system, and found that our sensor showed a remarkable current response for NO oxidation (10 μM), while the current responses from each of the potentially interfering species (10 μM) at the same potential are minimal, demonstrating the excellent selectivity of the sensor. This result was added in supporting information (Figure S10).

Reviewers' Comments:

Reviewer #1:

Remarks to the Author:

The novelty of this work has been better clarified and my other concerns have also been well addressed by the authors. I think the revised manuscript can be accepted for publication in Nature Communications.

Reviewer #2:

Remarks to the Author:

The authors addressed my concerns considerably. Therefore, I think this manuscript is publishable.

Reviewer #3:

Remarks to the Author:

The authors have revised the manuscript according to reviewers' comments and the significance of this work is well justified. This manuscript can be finally accepted. However, two issues are needed to be addressed.

1. For the synthesis of Ni NPs/N-C, the authors used the dopamine derived hollow carbon as precursor. If the authors used a similar synthetic procedure with Ni SACs/N-C in the presence of more metal salt, the Ni NPs/N-C can be obtained. Did the authors try this methods?
2. I do not think the characterization of Ni NPs/N-C is sufficient. For example, XRD pattern should be added. Ni XPS spectrum should be provided. For more accurate demonstration, Ni nanoparticles as a model is recommended to be investigated to probe the advantage of the Ni SACs.

Response to Reviewer #3

The authors have revised the manuscript according to reviewers' comments and the significance of this work is well justified. This manuscript can be finally accepted. However, two issues are needed to be addressed.

1. For the synthesis of Ni NPs/N-C, the authors used the dopamine derived hollow carbon as precursor. If the authors used a similar synthetic procedure with Ni SACs/N-C in the presence of more metal salt, the Ni NPs/N-C can be obtained. Did the authors try this methods?

Response: We thank this reviewer for the great comment. We did not try this method. This is because if we prepare the Ni NPs/N-C in a similar synthetic procedure to that of preparation of Ni SACs/N-C in the presence of more metal salt, it is highly possible that a mixture of Ni SACs and Ni NPs would be produced. To avoid yielding of the mixtures, researchers usually mix metal salts with suitable supporting materials, and then calcinate to obtain metal NPs as excellently demonstrated in the following literatures (*J. Am. Chem. Soc.* **2018**, *140*, 16936; *Nat. Commun.* **2018**, *9*, 2353).

2. I do not think the characterization of Ni NPs/N-C is sufficient. For example, XRD pattern should be added. Ni XPS spectrum should be provided. For more accurate demonstration, Ni nanoparticles as a model is recommended to be investigated to probe the advantage of the Ni SACs.

Response: We have included the XRD and XPS data of Ni NPs/N-C in Supplementary Figure 2, and Supplementary Figure 5a, respectively. Due to the low nickel content of 0.85% (wt%) of Ni in Ni NPs/N-C and the low resolution of XRD for the trace phase, no obvious Ni signal could be identified in XRD spectrum. However, the XPS results do reveal the presence of Ni in the Ni NPs/N-C.